# CT Angiography-Guided Needle Insertion for Interstitial Brachytherapy in Locally Advanced Cervical Cancer

**DOI:** 10.3390/diagnostics14121267

**Published:** 2024-06-15

**Authors:** Alexandra Timea Kirsch-Mangu, Diana Cristina Pop, Alexandru Tipcu, Alexandra Ioana Andries, Gina Iulia Pasca, Zsolt Fekete, Andrei Roman, Alexandru Irimie, Claudia Ordeanu

**Affiliations:** 1Department of Oncology, “Iuliu Hațieganu” University of Medicine and Pharmacy, 400347 Cluj-Napoca, Romania; timeakirsch@gmail.com (A.T.K.-M.); alexandru.tipcu@elearn.umfcluj.ro (A.T.); airimie@umfcluj.ro (A.I.); 2“Prof. Dr. I. Chiricuță” Oncology Institute, 400015 Cluj-Napoca, Romania; patcasdiana@yahoo.com (D.C.P.); sanda_andries@yahoo.com (A.I.A.); pasca_gina@yahoo.com (G.I.P.); claudia_ordeanu@yahoo.com (C.O.)

**Keywords:** CT angiography, cervical cancer, interstitial brachytherapy, needle insertion, uterine artery, fatal hemorrhage

## Abstract

CT angiography might be a suitable procedure to avoid arterial puncture in combined intracavitary and interstitial brachytherapy for cervical cancer curatively treated with combined chemoradiation and brachytherapy boost. Data in the literature about this technique are scarce. We introduced this method and collected brachytherapy data from patients treated in our department between May 2021 and April 2024. We analyzed the applicator subtype, needle insertion (planned versus implanted), implanted depth and the role of CT angiography in selecting needle trajectories and insertion depths. None of the patients managed through this protocol experienced atrial puncture and consequent hemorrhage. Needle positions were accurately selected with the aid of CT angiography with proper coverage of brachytherapy targets and avoidance of organs at risk. CT angiography is a promising method for guiding needle insertion during interstitial brachytherapy.

## 1. Introduction

Cervical cancer is both the fourth most common cancer and the fourth leading cause of cancer death in women worldwide, with 85% of reported cases occurring in developing countries according to Globocan 2022 [1]. External beam radiotherapy (EBRT) combined with concurrent platinum-based chemotherapy and high-dose-rate (HDR) brachytherapy (BT) boost is the standard treatment for locally advanced cervical cancer, in combination or as an alternative to surgery. Brachytherapy has become a crucial component of modern radiotherapy for locally advanced cervical cancers; it encompasses intracavitary brachytherapy (ICBT), interstitial brachytherapy (ISBT) and combined intracavitary/interstitial brachytherapy (IC/ISBT).

The addition of ISBT to ICBT increases the 3-year local control rate by 10%, without a significant increase in toxicity, especially in patients with larger, high-risk clinical target volume (HR-CTV > 30 cm^3^) [2]. ISBT not only enables adequate dose coverage of the parametrial extension of the tumor but also surmounts topography-related problems of the cervical primary. Nonetheless, the insertion of needles always remains a challenge due to potential arterial and bowel punctures.

At this time, there are no international guidelines for the optimal technique of needle insertion during ISBT for cervical cancer. In our study, we aimed to investigate the clinical feasibility and added value of computed tomography angiography (CTA) guidance (CTAG) to optimize needle insertion in order to avoid uterine artery (UA) puncture and massive hemorrhagic adverse events.

The uterine artery (UA), which supplies most of the blood to the uterus, is a major branch of the internal iliac artery. Understanding its anatomy and variations is crucial when performing invasive procedures.

The uterine artery follows a lateral-to-medial course through the lower portion of the broad ligament, also known as the cardinal ligament. At the level of the uterine isthmus, the uterine artery bifurcates into ascending and descending branches [3].

The influence of different anatomical variants of the UA to ISBT is unknown. Currently, there is a five-group classification of the uterine artery by origin: type I, 85% (origin of the UA is directly from the anterior trunk of the internal iliac artery), type II, 7% (from the branch of the anterior trunk of the internal iliac artery, called the inferior gluteal/superior vesical artery), type III, 5% (the origin of the UA is independent and comes directly from the internal iliac artery), type IV, 2% (from the umbilical branch of the anterior trunk of the internal iliac artery) and type V, 0.5% (origin directly from the ovarian artery, which in turn came from the left renal artery, instead of the abdominal aorta, as classically described) [4,5]. Albulescu et al. [6] conducted a study in which the origin and trajectory of the uterine artery were analyzed with 110 angiographies, resulting in a classification of four types, similar to those above. 

Furthermore, the clinical importance of understanding the UA variations lies in the fact that during salvage procedures, such as in uterine artery embolization, the anatomic variations can make the procedures quite difficult. Arterial embolization is considered today as the first-line therapy to control UA hemorrhage due to its quickness, excellent efficacy, minimal invasiveness, and uterine preservation [7].

During ISBT the operator can puncture the uterine artery or other branches of the internal iliac (or hypogastric) artery, which can result in immediate bleeding, or bleeding during needle removal. The resulting bleeding can sometimes be managed with local compression (tamponade with gauze), or it can require emergency surgery or embolization. Aggarwal et al. reported the first case of vascular rupture managed with local, intravascular embolization, which is a minimally invasive, efficient procedure [8].

The aim of our research was to assess the benefit of using CTAG in order to reduce the probability of puncturing any of the organs-at-risk, to evaluate different dosimetry implications of our ISBT method, along with a comparison of some of the available ISBT applicators currently on the market.

To the best of our knowledge, the precise relationship of the UA with the trajectory of the needles through the paracervical area on CT-based anatomy has not been previously described.

## 2. Materials and Methods

### 2.1. Patients

We collected brachytherapy data in a retrospective manner from patients treated in our department between May 2021 and April 2024. As selection criteria, we used interstitial brachytherapy with curative intent for cervical cancer patients who received CTAG before needle insertion and MRI and CT scan for planning. CTAG was introduced in 2019 in our treatment protocol for cervical cancer, following discussions on the gynecological multidisciplinary team meetings and its use was approved by the managerial board.

We included in our analysis data regarding demographics, histology, tumor grade, stage, dosimetry data (prescribed dose, EQD2, D90% to HR-CTV and needle insertion parameters, such as planned and actual inserted number of needles, along with minimum and maximum insertion depth. The staging was performed according to 2018 FIGO staging classification for cervical cancer. 

The ICBT insertion was performed using one of the following brachytherapy applicators: (1) a Vienna-style interstitial ring applicator (titanium tandem and ring tandem with a 7.5 mm plastic build-up cap), (2) an Aarhus Interstitial Ring applicator or (3) an Interstitial Universal Cylinder, all from Varian Medical Systems.

All patients received a CTAG scan at Siemens Somatom Confidence RT scanner, followed by, after needle insertion, a planning CT at 1.25 mm slice-thickness on a General Electric Discovery RT scanner and a 3 mm slice-thickness MRI scan on the Siemens 1.5 T Magnetom Aera scanner. For image fusion, the MRI was registered to the planning CT using automated rigid registration on bone structures, continued by manual matching with focus on the applicator and soft tissue in the immediately adjacent area.

We contoured the HR-CTV on the T2-weighted oblique MRI series, with slices angled parallel to the ring and cylinder transverse plane. Contouring of organs-at-risk (OAR) and reconstruction of the applicator was performed on the planning CT images. The applicator was reconstructed with the help of the solid applicator library.

Treatment plans were created in Varian BrachyVision version 17.0 from Varian Medical Systems. For calculations, we used the nominal source strength of 10 Ci 40,700 cGy cm^2^/h) for 192 Ir and we applied a TG-43 line-source dose calculation formalism. At the treatment console, the treatment time was recalculated automatically using the actual source strength at that time.

### 2.2. Implant Procedure and Image Acquisition

Based on the pre-EBRT-MRI, we assessed the residual parametrial involvement and we estimated the optimal number of needles to be inserted.

The first step in the brachytherapy procedure is to prepare the patient by insertion of a urinary catheter and voiding of the bladder. Next, we perform the insertion of intracavitary (IC) applicator under sedation with nitrous oxide, using a continuous flow-mixer for analgesia.

After the insertion of the IC applicator, we transfer the patient to the CT room to undergo CTAG. With the bolus trigger region of interest at the aortic bifurcation, 100 mL iodinated contrast agent is injected at a rate of 4 mL/s using a 6 s delay. The scan is performed craniocaudally from 5 cm above the umbilicus to the minor femoral trochanter using a 120 kV 350–450 mA, table pitch 1, zero gantry pitch protocol, with automatic tube current modulation and 0.6 mm collimation width. Images are reconstructed with a slice thickness of 1 mm. 

Afterward, the CTAG images are reviewed on a dedicated workstation by a brachytherapy medical physicist and a radiologist with the purpose of identifying the anatomical structures along the needle’s paths. Multiplanar reconstructions (MPR) are created along the last 5 mm of each needle channel axis, to account for the distal angles of the three types of applicators (Figure 1). We measured the distance between the upper orifice of each channel and organs-at-risk predisposed to puncture complications, such as the uterine arteries, ureters, urinary bladder, rectum and bowel (Figure 2). Based on these measurements, optimal needle insertion depth is calculated, while keeping a safety distance of 5 mm from the organs previously mentioned.

During the needle assessment on the CTAG images we also completed a standardized form to account for the in-use applicator channels. The form serves both the current insertion and the accurate documentation of the treated channels.

Following the needle insertion and CTAG assessment, after the patient recovery from sedation, we proceed with the planning CT and MRI image acquisition, used for contouring the target volumes and the organs-at-risk, and dose calculation.

The first assessment of the possible hemorrhagic complications due to the needle insertion occurs when we analyze the CT and MRI images.

### 2.3. Contouring for Brachytherapy: OARs, GTV Res, Adaptive HR-CTV, IR-CTV

Contouring, dose coverage and dose constraint values were assessed according to the standards set by the EMBRACE II Study [9].

Both tumor and OAR contouring are performed for each individual insertion/implant of the BT applicators. Target volumes are contoured on T2 weighted transversal MRI sequence in a dedicated 3D brachytherapy dose-planning system according to the GEC ESTRO recommendations and the ICRU/GEC ESTRO report 89. The MRI-based target delineation can be reused by superimposition in the process of contouring on CT if for subsequent fractions only a planning CT is performed with the applicator in place.

The following targets are contoured: GTV res: residual (hyperintense) Gross Tumor Volume of the primary Tumor;HR-CTV: adaptive High-Risk Clinical Target Volume of the primary Tumor;IR-CTV: intermediate Risk Clinical Target Volume of the primary Tumor.

The following organs are contoured (from at least 2 cm below the IR-CTV to 2 cm above the uterus):Bladder: outer bladder wall including the bladder neck;Rectum: outer rectal wall from the anal sphincter to the transition into the sigmoid;Sigmoid: outer sigmoid wall from the recto-sigmoid flexure to at least 2 cm above the parametria and the uterus;Bowel loops: outer contours of loops positioned within 3–4 cm of the uterus and applicator.

After the contouring is approved by the attending radiation oncologist, the contours from the MRI are transferred to the CT images used for treatment planning.

### 2.4. Planning Aims and Dose Prescription

Our planning aim was that the dose received by 90% of the HR-CTV volume (D90% HR-CTV) should be between 90 and 95 Gy (combining the dose received in external beam radiotherapy and BT dose), while keeping the 2 cm^3^ dose (D_2_cm^3^) for bladder below 80 Gy, D_2_cm^3^ for rectum below 65 Gy, D_2_cm^3^ for sigmoid/bowel below 70 Gy (soft constraints based on EMBRACE II). Taking into account the individual patient anatomy, geometric possibilities of the applicators and dose optimization, deviations from these planning aims are allowed. The tolerance dose for the D90% HR-CTV > 85 Gy. Tolerance limits for OARs: D_2_cm^3^ for bladder < 90 Gy, D_2_cm^3^ for rectum < 75 Gy and D_2_cm^3^ for sigmoid/bowel < 75 Gy (limits for prescribed dose, or hard constraints). Deviations from these constraints are only allowed in special cases with detailed explanations/motivations. For OARs, there are also two levels with planning aims 5–10 Gy lower than the maximum limits for the prescribed dose.

Planning aims (soft constraints) and limits for prescribed dose (hard constraints) for treatment planning extracted from EMBRACE II can be visualized in Table 1. The EQD2 is calculated using an alfa/beta = 10 for targets, alfa/beta = 3 for OARs (late effects) and a repair half-time of 1.5 h. Doses were calculated using the BT spreadsheet created by Nag and Gupta in 1998 [10].

### 2.5. Brachytherapy Workflow and Patients

In order to reduce adverse reactions due to sedation and for better patient workflow, in our brachytherapy roadmap we deliver the first fraction on the day of the procedure (needle insertion), followed by another fraction the following morning. As such, the patient remains with the applicator in situ overnight.

After the first treatment delivery, patients are admitted into the inpatient ward with the urinary catheter left in place. The nursing staff records any signs of active bleeding or hematuria. Intravenous pain relievers are given for pain control from the moment of needle insertion up to patient discharge. Patients also receive hydric diet and loperamide for peristalsis reduction [11]. A minimum time of 6 h between each HDR brachytherapy fraction is respected (from the first to second treatment there are 18 h and from the second to the third there is one week).

Before the second treatment delivery, the following morning, a new planning CT scan is acquired for the evaluation of the implant position relative to the prior day. Fusion with the first planning CT scan is performed, followed again with the rigid automated image fusion on bone structures and then manual image matching based on BT applicator. We evaluate the contouring from the first day CT scan duplicated on the second day CT images. According to our institutional procedure, if there is a difference in needle depth length or position movement greater than 3 mm, then re-contouring is carried out, otherwise, we proceed to treatment with the same treatment plan from the day before.

After the second BT fraction delivery, the applicator is removed, and the patient is discharged as long as there is no active bleeding present. Patients receiving a subsequent applicator implant are readmitted the following week and similar care is provided. Our clinical workflow can be observed in Figure 3.

### 2.6. Statistical Analysis

Data collection was performed using Microsoft Office 365 Suite—Office Excel. Data processing, statistical analysis and chart generation were performed using International Business Machines Corporation^®^ Statistical Product and Service Solutions^®^ v.26.0.0. 

Data distribution assessment was performed using Kolmogorov–Smirnov (with Lilliefors significance correction) and Shapiro–Wilk tests, along with the distribution plots. Independent groups were compared using One-Way ANOVA (post hoc Tukey HSD) or Kruskal–Wallis H (post hoc Dunn–Bonferroni), based on variable distribution. Frequency comparison and testing for independence were performed using Chi-square statistics. Linear relationships were assessed by Spearman correlations. Multinomial logistic regressions were performed in order to estimate the independent influence of each co-dependent nominal variable. All tests were considered to be statistically significant at an alpha of 0.05, while also adjusting for multiplicity.

## 3. Results

In the selected period, we identified 44 locally advanced cervical cancer patients treated with definitive chemo-radiotherapy (stage IB2 through IIIC2). 

Most patients presented with squamous cell carcinoma (41 patients) with different grades of differentiation. Two patients presented papillary-squamous cell carcinoma, and one patient had clear cell adenocarcinoma. 

The age of the patients ranged from 31 to 81 years, with a mean of 51.32 ± 23.87 years.

Patients received a total dose of 45 Gy to 50.8 Gy through image-guided volumetric modulated arc therapy (IG-VMAT) with or without a simultaneous integrated boost for macroscopical enlarged lymph nodes (55–57.5 Gy/25), and concomitant chemotherapy with Cisplatin at a dose 40 mg/m^2^ once a week for a total of five–six cycles.

EBRT was followed by HDR ISBT brachytherapy, most commonly with 28 Gy in four fractions (14 patients) or less with 24 Gy in four fractions (23 patients), 21 Gy in three fractions (6 patients) and 14 Gy in two fractions (1 patient), depending on the IG-VMAT dose and on the D90% HR-CTV EQD2 dose.

Patient characteristics are shown in Table 2.

The EBRT dose was established by each physician based on the clinical risk factors of the patients. The dose prescription ranged from 45 Gy/25 (33 patients) to 50.4/28 fractions (9 patients) with two patients having a different dose regimen with or without simultaneous integrated boost to the pathological lymph nodes. Of the two patients, one was treated with hemostatic intent due to massive hemorrhage at diagnosis, and one was treated with sequential boost after the primary EBRT dose, both in a secondary clinic.

Based on the initial tumor extent at diagnosis, the appropriate IC applicator was chosen in order to perform the brachytherapy sessions (Figure 4).

All patients achieved a D90% to HR-CTV of 90 to 95 Gy (EQD2). The majority (56.81%) of the patients reached between 92 Gy and 95 Gy of the equivalent dose, while the rest had an equivalent of 90 Gy to 92 Gy. There was no statistically significant difference between the three applicator types with regard to D90% to HR-CTV D90% (*p* = 0.688).

In our study, there was no instance of hematoma or injury to the surrounding organs during the brachytherapy procedure for any of the treated patients.

The number and position of needles used depended on the proximity of the UA and OARs. The positions at risk for puncture were avoided, or needles were inserted to a certain limited depth. 

We compared the three types of applicators regarding needles “lost” between theoretical planning (of needle positions and lengths) and the actual needle insertion, based on the CTAG. Because the total number of needle insertions differs for each applicator type, we expressed the number of “lost” needles as a percentage of the total planned needles, based on prior clinical and imaging data. The Vienna ring lost on average 26.42% of the needles, followed by the Aarhus ring with 18.57%, and then the interstitial cylinder, with an average of 9.86% (*p* = 0.015). These differences in the “lost” needles can be observed in Figure 5. With regard to insertion range (defined as the difference between the deepest and the most superficial needle insertion depth), the differences between the three applicators did not reach statistical significance (averages of 11 mm for the cylinder, 8.75 mm for the Vienna ring and 14.38 mm for the Aarhus ring), *p* = 0.116.

The number of needles initially planned versus actually inserted can be observed in Figure 6. The average number of the needles “lost” when we used the interstitial cylinder was 0.66 (from 6.35 planned to 5.65 inserted), 1.56 (from 5.69 planned to 4.13 inserted) for the Vienna applicator, and 2.62 (from 12.25 planned to 9.63 inserted) for the Aarhus applicator, *p* = NS.

Out of the four possible limiting factors for the number of inserted needles, the most prevalent was the bladder anatomy in relationship with the applicator (36.36% of cases), followed by uterine artery anatomy (34.09%), rectum anatomy (15.91%) and lastly, bowel anatomy (2.27% of cases) (Figure 7).

The applicator with the most limiting factors regarding the number of inserted needles was the Vienna ring (avg 1.19), followed by the Aarhus ring (avg 1.13), and lastly the interstitial cylinder (avg 0.55) (*p* = 0.042) (Figure 8).

As for insertion depth, the bladder anatomy interfered in 50% of cases while the uterine artery interfered in 54.55% of cases. There were no statistically significant differences between the three applicators when compared by the number of present limiting factors with respect to insertion depth (*p* = 0.334) (Figure 9).

When comparing each individual limiting factor, for each of the two aspects (number of needles and insertion depth), we found that the bladder interfered, with respect to the number of needles, more frequently in the case of the Vienna ring, compared to the interstitial cylinder (OR = 8.546, CI 95% 1.647–44.341, *p* = 0.011). No other significant differences were found between the limiting factors.

There was no statistically significant association between disease stage and applicator type (*p* = 0.126). We found no statistically significant linear correlation between the percentage of non-viable needles and insertion depth range (rho = 0.165, *p* = 0.284).

Finally, we report the CTDI (CT dose index) for CTAG (CT angiography) in Table 3, for all 44 patients. As we described above, the examination is a double (both native and contrast) abdominal and pelvic scan.

## 4. Discussion

We treated 44 locally advanced cervical cancer patients (stage IB2 through IIIC2) with definitive chemo-radiotherapy with an HDR BT boost, guided by CT angiography.

MRI provides excellent soft-tissue contrast and allows for the individualization of the radiation dose to the HR-CTV and minimizes the dose to the OARs [12]. In addition to MRI a CT angiography can provide better visualization of the uterine artery and other arterial branches. We think that this extra step can make interstitial brachytherapy for cervical cancer safer. While the literature supports the use of transrectal ultrasound for guiding needle insertions [9], this technique remains less accessible due to the requirement for specialized training. Additionally, most radiation oncologists lack proficiency in this method. Furthermore, the ultrasound systems employed for interstitial brachytherapy are specialized robotic arm systems, which are not universally available across all hospitals.

The advantage of ultrasound guidance consists of the real-time adjustment possibilities it can offer [13] and the lack of exposure to ionizing radiation.

Additional CT imaging adds additional exposure to ionizing radiation. CTAG added an additional 38.1 millisieverts of radiation on average to the dose received by the patient. There is always a chance of secondary cancer from exposure to radiation. However, the benefit of an accurate diagnosis far outweighs the risks involved with CT scanning. 

In a recent meta-analysis conducted by Cao et al. [14], it was found that adults who undergo CT scans face an increased risk of cancer. Specifically, the odds ratio (OR) for cancer risk following CT scans was 5.89 (95% CI: 3.46 to 10.35) when compared to non-exposed individuals. This risk was positively correlated with both radiation dose and the number of CT scan sites. Notably, the risk of solid malignancy was higher than that of leukemia. Interestingly, the risk was statistically significant only in subjects less than 45 years of age.

On the other hand, another meta-analysis suggests that exposure to multiple CT scans and other sources of low-dose radiation (up to 100 mSv or approximately 10 scans, or even 200 mSv) does not significantly increase cancer risk [15].

The cure rate of advanced cervical cancer is high, at approximately 60–85% (depending on the stage), thus survival of these patients might frequently exceed 20–30 years, considering the medium age at presentation of 45–55 years. Radio-induced cancers usually occur more frequently at around 10–20 years after exposure (for solid tumors) and at 5–10 years for hematological malignancies (with extremes at 1 year to more than 60 years) [16,17,18], and thus radiation-induced cancers could be observed in patients treated with cervical cancer. But even in the case of higher doses delivered during radiotherapy, the risk of radio-induced cancers appears to be modest. In the retrospective study of Rombouts et al. [19], the OR for rectal cancer after RT for endometrial cancer was only 1.50, 95% CI: 1.13–2.0. 

Reference centers for cervical BT should implement either CT or US guidance when employing interstitial BT.

In certain cases, contraindications for administering iodinated contrast can pose challenges. For patients with such contraindications, it is advisable to consider alternative imaging methods for guidance. Such contraindications are allergy to iodinated contrast agents and reduced glomerular filtration rate. In situations where allergic reactions are a concern, steroids may be employed to mitigate the risk. However, it is essential to recognize that this approach carries significant potential risks.

Another drawback when adopting CTAG could be that not all radiotherapy departments are equipped with automated contrast injectors.

Avoidance of arterial or organ punctures could be avoided in all our cases.

The blood flow through the uterine arteries is not extremely high [20], although not negligible, and is around 100 mL/mL [21]. Its puncture may result in rapid bleeding if not promptly recognized. The blood flow in the uterine arteries increases with previous pregnancies [22,23].

The experience in UA catheterization and embolization mainly comes from the management of perinatal rupture of the uterine arteries. The blood loss in these situations is much more severe than in non-pregnant women [24,25,26]. Laparoscopic or open surgery is an alternative method to stop the bleeding [27].

The number of needles “lost” was higher for the cylinder than for the ring applicators. This can be explained by the limited number of needle channels available in comparison to the ring applicators.

There is no description in the literature of CTAG to aid needle insertion during interstitial brachytherapy. Even the information regarding the standalone CTAG procedure in cases of uterine malignancies remains limited. Fujioka et al. [28] give a detailed account use of preoperative CTAG of UA in nine patients with cervical cancer. Their primary goal was to guide surgery to avoid complications when the crossing site of the UA and ureter is dissected. 

CTAG is not regarded as a standard examination when evaluating cervical cancer [29]. However, CTAG was evaluated to describe the relationship of lymph node metastases to big vessels [30]. 

Although the sample size and our experience are limited, none of our patients experienced hemorrhagic complications due to needle insertion with the aid of computed tomography angiography. 

From our experience, we consider that CTAG is a feasible safety measure in the planning of interstitial brachytherapy for locally advanced cervical cancer, especially when robotic arm ultrasound dedicated to brachytherapy is not available. Comparison with ultrasound guidance merits further investigation. MR angiography should also be tested, although it has an inferior resolution compared to CTAG [28].

## Figures and Tables

**Figure 1 diagnostics-14-01267-f001:**
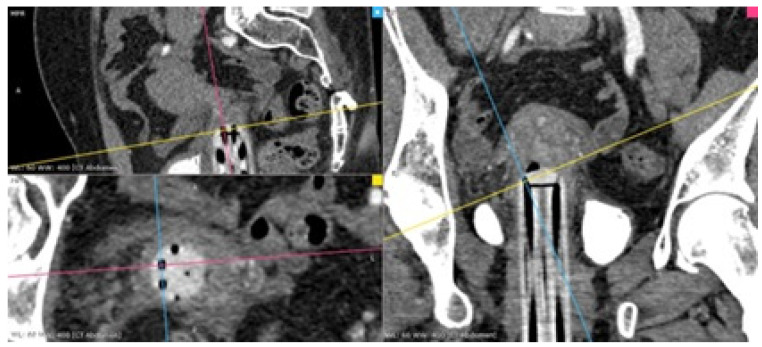
MPR reconstruction along the 10 o’clock channel of a Baldovin Maria device. The images are reconstructed taking the angulation of the channel into account. (P-posterior, R-right, colored squares: image corner).

**Figure 2 diagnostics-14-01267-f002:**
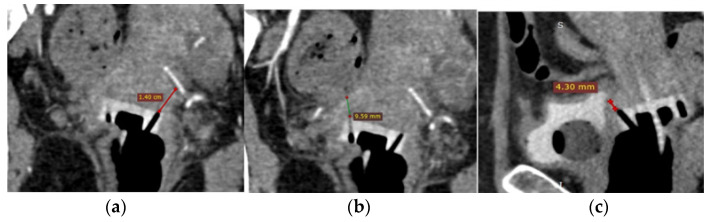
Distances between the internal channel orifices of an Aarhus ring device and (**a**) the uterine artery, (**b**) the sigmoid colon, (**c**) the bladder. (S-sagittal).

**Figure 3 diagnostics-14-01267-f003:**
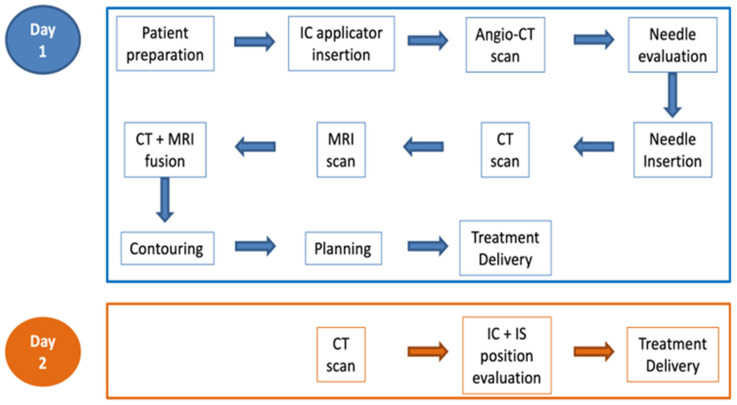
ISBT clinical workflow.

**Figure 4 diagnostics-14-01267-f004:**
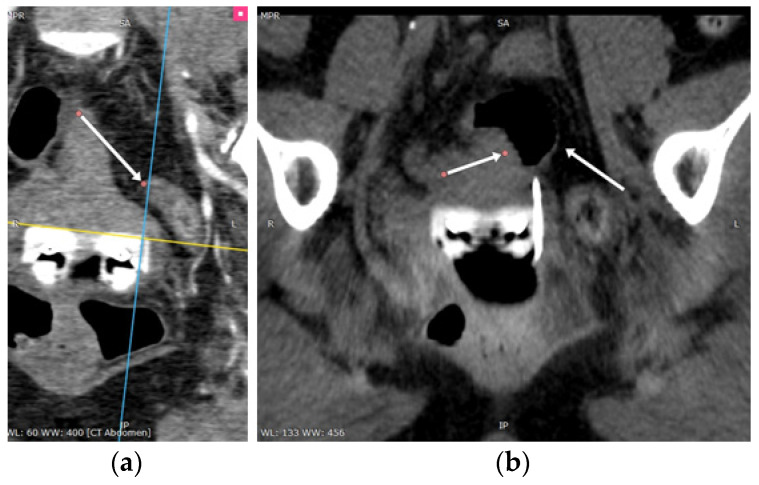
(**a**) MPR reconstruction along the 6 o’clock channel of Vienna ring device. The sigmoid colon can be seen at 10 mm in front of the internal opening. (**b**) Post-insertion scan of the same patient shows the tip of the applicator in proximity of the bowel, without penetrating it. (All arrows point to bowel.)

**Figure 5 diagnostics-14-01267-f005:**
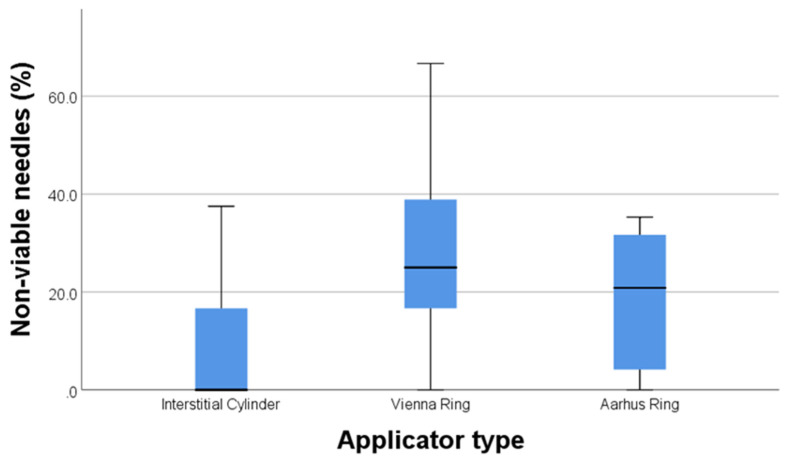
Percentage of needles “lost” after adjusting for needle insertion planning based on CTAG.

**Figure 6 diagnostics-14-01267-f006:**
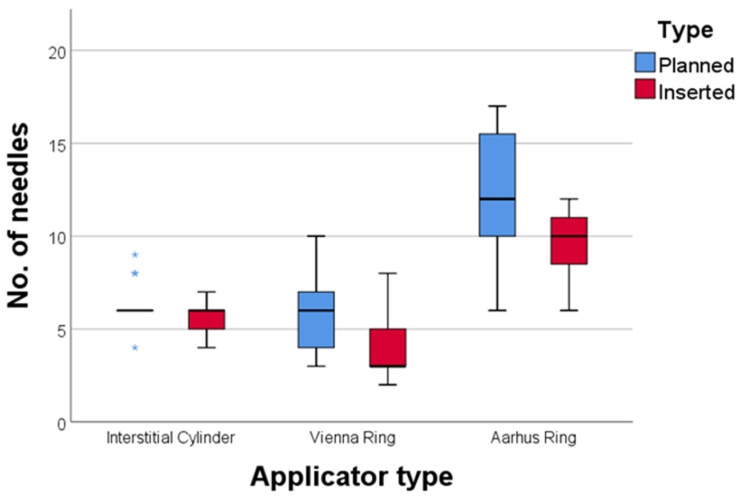
Distribution of the number of the planned versus inserted needles for each applicator type (* extreme outlier values).

**Figure 7 diagnostics-14-01267-f007:**
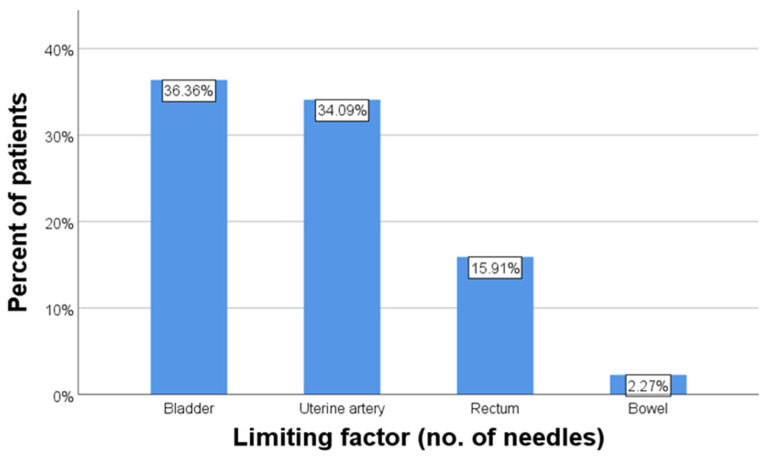
Limiting factors for needle insertion.

**Figure 8 diagnostics-14-01267-f008:**
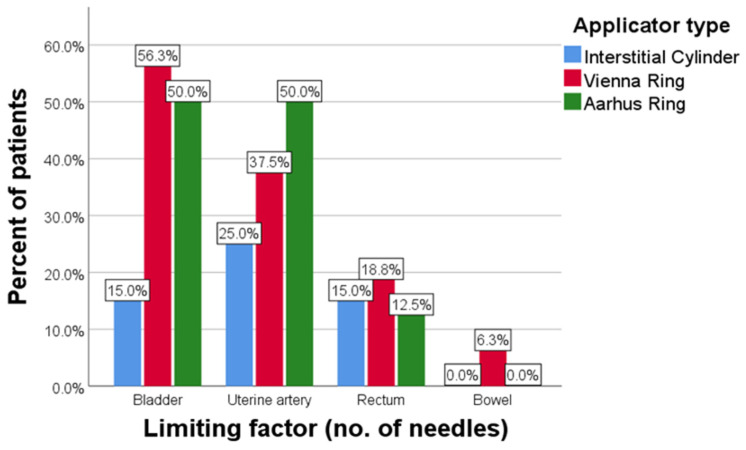
Limiting factors for needle insertion by applicator type.

**Figure 9 diagnostics-14-01267-f009:**
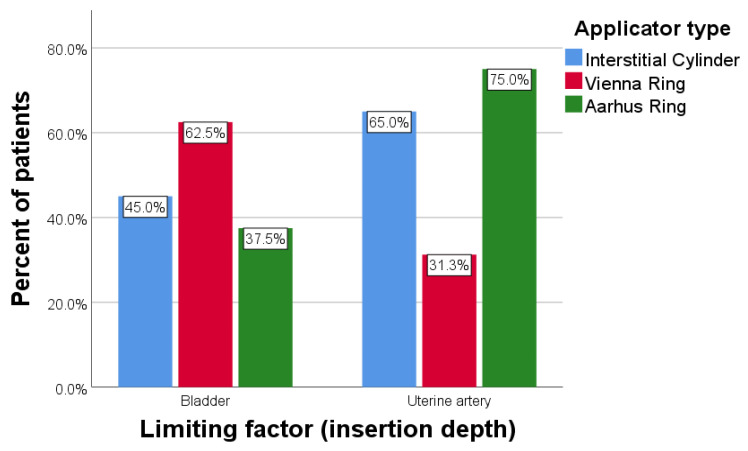
Limiting factors for needle insertion depth by applicator type.

**Table 1 diagnostics-14-01267-t001:** Planning aims (soft constraints) and limits for prescribed dose (hard constraints) for treatment planning, extracted from EMBRACE II study.

**Target**	**D90 CTV_HR_** **EQD2_10_**	**D98 CTV_HR_** **EQD2_10_**	**D98 GTV_res_** **EQD2_10_**	**D98 CTV_IR_** **EQD2_10_**	**Point A** **EQD2_10_**
**Planning aims**	>90 Gy <95 Gy	>75 Gy	>95 Gy	>60 Gy	>65 Gy
**Limits for prescribed dose**	>85 Gy	-	>90 Gy	-	-
**OAR**	**Bladder D_2cm_^3^** **EQD2_3_**	**Rectum D_2cm_^3^** **EQD2_3_**	**Recto-vaginal point** **EQD2_3_**	**Sigmoid D_2cm_^3^** **EQD2_3_**	**Bowel D_2cm_^3^** **EQD2_3_**
**Planning aims**	<80 Gy	<65 Gy	<65 Gy	<70 Gy	<70 Gy
**Limits for prescribed dose**	<90 Gy	<75 Gy	<75 Gy	<75 Gy	<75 Gy

**Table 2 diagnostics-14-01267-t002:** Patient characteristics.

Variable	Classifier	Frequency	(%)
Stage (FIGO)	IB2	2	4.76
	IB3	1	2.8
	IIA	1	2.38
	IIB	9	19.05
	IIIA	1	2.38
	IIIB	7	16.67
	IIIC1	18	40.48
	IIIC2	4	9.52
	IVA	1	2.38
Applicator type	Universal interstitial cylinder	20	45.5
	Aarhus ring	8	18.2
	Vienna ring	16	36.3
EBRT dose	45 Gy/25 fr	33	75.0
	50.4 Gy/28 fr	9	20.4
	45 Gy/25 ft + 16 Gy/8 fr	1	2.3
	30 Gy/20 + 24 Gy/12	1	2.3
BT dose	4 × 7 Gy	14	31.8
	4 × 6 Gy	23	52.3
	3 × 7 Gy	6	13.6
	2 × 7 Gy	1	2.3
SCC Grading	G1	1	2.3
	G2	22	50.0
	G3	18	40.9
Other histology	Clear Cell Adenocarcinoma	1	2.3
	Papillary Squamous	2	4.5

**Table 3 diagnostics-14-01267-t003:** CTDI (CT dose index) for CTA (CT angiography) for our patients.

Patient	Dose in mSv (Millisievert)
**1**	27.65
**2**	38.76
**3**	51.49
**4**	26.6
**5**	50.01
**6**	21.64
**7**	34.57
**8**	49.37
**9**	38.03
**10**	36.11
**11**	32.37
**12**	34.69
**13**	45.66
**14**	40.27
**15**	22.18
**16**	31
**17**	27
**18**	26.5
**19**	50
**20**	36.36
**21**	62.38
**22**	30.12
**23**	33.56
**24**	33.1
**25**	39.11
**26**	29.8
**27**	28.67
**28**	55.98
**29**	39.82
**30**	43.22
**31**	24.59
**32**	39.8
**33**	39.06
**34**	20.8
**35**	72.28
**36**	34.1
**37**	38.6
**38**	47.64
**39**	31.41
**40**	37.44
**41**	49.04
**42**	62.76
**43**	38.8
**44**	21.93
**Average**	**38.1**

## Data Availability

The data presented in this study are available on request from the corresponding author. The data are not publicly available due to patient privacy.

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
