# Peer review of "CT Angiography-Guided Needle Insertion for Interstitial Brachytherapy in Locally Advanced Cervical Cancer"

_diagnostics, 2024, doi:10.3390/diagnostics14121267_

Round 1

Reviewer 1 Report

Comments and Suggestions for Authors

The authors presented the study entitled” CT-angiography guided needle insertion for interstitial brachytherapy in locally advanced cervical cancer” by including 44 locally advanced cervical cancer patients treated with definitive chemo-radiotherapy. An idea developed and verified for locally advanced cervical cancer patients in interstitial brachytherapy.

In material and method section, patient characteristics (or including criteria), does comorbidity such as hypertension or some medical situations concerned for the procedure step of CT-angiography.

In discussion section, as the cases received the brachytherapy between May 2021 and April 2024, it is a limitation for analyzing prognosis in the not long period. It is necessary to exam the prognosis for the CTAG than those without the CT-angiography guided.

Comments on the Quality of English Language

Minor editing of English language required

Author Response

Dear Reviewer, Dear Colleague,

We appreciate your time and dedication to thoroughly analyze our article. We have tried as much as we could to address your questions/concerns. Please find our responses inline. We have edited our manuscript as well.

  1. In material and method section, patient characteristics (or including criteria), does comorbidity such as hypertension or some medical situations concerned for the procedure step of CT-angiography?

Thank you for your suggestion. We have added contraindications for CT-angiography. We have analyzed only patients, who could perform the procedure.

We have added the following text to the Discussion section:

In certain cases, contraindications for administering iodinated contrast can pose challenges. For patients with such contraindications, it’s advisable to consider alternative imaging methods for guidance. Such contraindications are allergy to iodinated contrast agents and reduced glomerular filtration rate. In situations where allergic reactions are a concern, steroids may be employed to mitigate the risk. However, it’s essential to recognize that this approach carries significant potential risks.

  1. In the discussion section, the cases received the brachytherapy between May 2021 and April 2024, it is a limitation for analyzing prognosis in the not long period. It is necessary to exam the prognosis for the CTAG than those without the CT-angiography guided.

I absolutely understand your concern. Our goal was to analyze the safety and feasibility of the procedure. The procedure is new, but all other procedures followed institutional protocol. Most importantly, the dose coverage of the target was not altered with the use of CT-angiography. We succeeded in covering the target with the inserted needles.

Our goal this time was not to report treatment results. We would like to publish our results in a separate article in the following years.

We appreciate your valuable input once more!

Dr Fekete

Reviewer 2 Report

Comments and Suggestions for Authors

The article is prepared with the claim of introducing a new methodology to a lesser-known topic. The study design is appropriate for the objective, and the findings are presented in a sufficient level of detail in an organized manner. However, some shortcomings are noticeable:

1. Abstract-conclusion: The concluding sentence is too assertive. Instead, the phrase "promising method" would be more appropriate.

2. The study describes a CT-based method. Therefore, the X-ray dose information should be provided. The CTDI value of the dose delivered in each examination and the total average value should be presented, and in the discussion section, the potential X-ray-related harms and the delivered dose should be discussed.

3. The discussion section is too short and inadequate:

   - The first paragraph of the discussion should briefly summarize the study.

   - General anatomical information should be placed earlier in the text.

   - The methods used in this procedure and their shortcomings should be described in more detail later.

   - Efforts to address these shortcomings, especially referencing recent studies, should be examined in detail.

Author Response

Dear Reviewer/ Dear Colleague,

Your thoughtful and constructive feedback has been received with enthusiasm. Our detailed responses are provided below, addressing each point directly, with inline text. The article has been modified accordingly.

The article is prepared with the claim of introducing a new methodology to a lesser-known topic. The study design is appropriate for the objective, and the findings are presented in a sufficient level of detail in an organized manner. However, some shortcomings are noticeable:

  1. Abstract-conclusion: The concluding sentence is too assertive. Instead, the phrase "promising method" would be more appropriate.

We have modified the conclusion to correct this issue.

  1. The study describes a CT-based method. Therefore, the X-ray dose information should be provided. The CTDI value of the dose delivered in each examination and the total average value should be presented, and in the discussion section, the potential X-ray-related harms and the delivered dose should be discussed.

We have added a table of the CTDI values (Table 3), expressed in mSv and discussed the related potential harms.

  1. The discussion section is too short and inadequate:

   - The first paragraph of the discussion should briefly summarize the study.

Thank you, we have corrected this aspect.

   - General anatomical information should be placed earlier in the text.

We have moved the anatomical information to the Introduction section.

 - The methods used in this procedure and their shortcomings should be described in more detail later.

   - Efforts to address these shortcomings, especially referencing recent studies, should be examined in detail.

Certainly! In our recent revision, we incorporated 18 additional references. Some of these specifically address angiography of the pelvis and cervical cancer. However, despite our thorough investigation, we found that the relationship between the uterine artery and the trajectory of needles through the paracervical area based on CT anatomy has not been previously documented. Unfortunately, our search for relevant publications encountered challenges. We provided supplementary information when appropriate.

Once more, we express our gratitude for your valuable time and insightful feedback.

Dr Fekete

Round 2

Reviewer 2 Report

Comments and Suggestions for Authors

The revised version of the article has reached an acceptable level, with all suggestions being satisfactorily addressed.